# Impact of Delirium-Related Stress, Self-Efficacy, Person-Centred Care on Delirium Nursing Performance Among Nurses in Trauma Intensive Care Units: A Cross-Sectional Descriptive Survey Study

**DOI:** 10.3390/healthcare13111243

**Published:** 2025-05-25

**Authors:** Ga-Hee Seong, Hyung-Ran Park

**Affiliations:** College of Nursing, Research Institute of Nursing Science, Chungbuk National University, Cheongju 28644, Chungbuk, Republic of Korea; sgh7985@gmail.com

**Keywords:** trauma intensive care unit, delirium, person-centred care, self-efficacy, delirium nursing performance

## Abstract

**Background/Objectives**: Enhancing delirium nursing performance in trauma intensive care units (TICUs), where the prevalence of delirium is high, causes early detection of delirium and improves the quality of nursing care. TICU nurses experience various stress levels while caring for patients with delirium, which negatively affects their performance. Self-efficacy improves delirium nursing performance based on their capacity. Person-centred care identifies the holistic needs of patients in TICUs, which stimulates their recovery. This study aimed to examine the relationship of delirium-related stress, self-efficacy, and person-centred care with delirium nursing performance and identify factors influencing delirium nursing performance among nurses in TICUs. **Methods**: This cross-sectional descriptive survey study was conducted on 170 TICU nurses from eight hospitals in Korea from 22 July to 30 September 2024. Data was collected using self-reported questionnaires after informed consent was provided. Data were analysed using multiple regression analysis. **Results**: Delirium nursing performance showed significant positive correlations with person-centred care (*r* = 0.51, *p* < 0.001) and self-efficacy (*r* = 0.41, *p* < 0.001). Regression analysis revealed person-centred care (β = 0.46, *p* < 0.001) and self-efficacy (*β* = 0.24, *p* = 0.004) as significant predictors of delirium nursing performance in TICUs, accounting for 28.6% of the variance. **Conclusions**: Interventions focused on person-centred care may help improve delirium nursing performance and practice holistic care.

## 1. Introduction

The annual number of patients with severe trauma is 194,927, with common mechanisms of injury including accidents, self-harm and suicide attempts, physical violence, and attempted homicide [1]. Most of these patients frequently experience delirium upon admission to a trauma intensive care unit (TICU) [2]. The incidence of delirium after cardiac surgery ranged from 4.1% to 54.9% [3] and was 32.5% in general intensive care units (ICUs) [4]. Meanwhile, the incidence of delirium in patients with severe trauma ranged from 50% to 75% [5]. Despite patients with trauma being typically younger [6] and having fewer comorbidities than patients with no trauma, the incidence of delirium is considerably higher in patients with trauma due to the nature of trauma-related injuries [7]. In TICUs, patients with multiple organ failure and massive haemorrhage requiring fluid therapy and blood transfusions are at particularly high risk for developing delirium. Because of additional risk factors such as fluid and electrolyte disorders, delirium is frequently observed in TICUs [8,9]. Delirium is associated with increased mortality, prolonged hospital stays and readmission, long-term cognitive impairment, and increased healthcare costs, ultimately impacting the post-treatment quality of life for patients and their families [10]. Therefore, targeted care is essential to reduce the risk of delirium in these patients.

Early diagnosis of delirium in critically ill patients with severe trauma is challenging due to the complexity of their conditions and the frequent use of sedation [11]. Nevertheless, TICU nurses play a critical role in closely monitoring patients’ conditions and delivering essential care. Prompt identification of delirium-related risks through vigilant care, defined as delirium nursing performance [12], is crucial for effective intervention.

Furthermore, identifying factors that positively influence delirium nursing performance is important [13]. Systematic literature reviews previously highlighted knowledge of delirium nursing performance as a key contributing factor [13,14]. More recently, research has broadened its focus to include emotional stress experienced by nurses [14,15,16,17], self-efficacy as an indicator of nursing competence in managing delirium [15,17,18,19], and person-centred care as a foundational element of holistic nursing practice [20,21].

According to a systematic review and qualitative study on nurses’ experiences of providing care for patients with delirium, nurses reported experiencing varying levels of stress and negative emotions while providing care for such patients [14]. Delirium-related stress is experienced by nurses during their care of patients with delirium [22]. Delirium-related stress negatively correlates with care quality among nurses working in recovery rooms [15]. Particularly, patients with delirium in ICUs often exhibit hyperactive delirium. Verbal threats and aggressive behaviours can lead to a decline in the quality of delirium care provided by nurses, who serve as primary healthcare providers [16]. Therefore, further research is warranted to investigate the relationship between nurses’ delirium-related stress and delirium nursing performance in TICUs.

Patients in ICUs often experience acute delirium, resulting in significant fluctuations in their clinical status. Therefore, early diagnosis and prompt intervention are among the greatest challenges experienced by ICU nurses when managing delirium [19]. One of the most critical predictors of effective and successful delirium nursing performance in ICUs is self-efficacy. Self-efficacy refers to nurses’ belief in their ability to perform care independently [11,17]. Nurses with high self-efficacy have confidence in their capabilities to manage clinical challenges based on their competence, which enhances their ability to provide effective care [17], ultimately contributing to improved patient safety and quality of nursing care [19]. Nurses with high post-operative self-efficacy demonstrated significantly better performance in delirium care [18]. Notably, the incidence of delirium is higher in patients in TICUs than in patients in general ICUs. Therefore, conducting research is essential to investigate the relationship between self-efficacy and delirium nursing performance among TICU nurses to enhance the quality of care.

Since most patients in TICUs have injuries severe enough to determine their survival, they are typically intubated with an endotracheal tube and sedated for intensive treatment and monitoring [20]. Their broader care needs are often overlooked, with treatment primarily focused on managing acute-phase injuries. Patients are restrained in bed and may experience derealisation, fear, and panic attacks due to limited mobility and lack of communication. These conditions can exacerbate the emotional and behavioural manifestations of delirium [21]. In such situations, patients’ recovery often depends on nurses’ sensitivity and competence in recognising and addressing their holistic needs [20,23]. Person-centred care is a holistic nursing approach focusing on the individual needs of patients, respecting their personal choices, and protecting their autonomy and dignity [24,25,26]. According to the conceptual framework of person-centred care, effective care delivery requires that the healthcare system must understand and meet patients’ needs [27]. Accordingly, delirium nursing performance in TICUs can be improved by applying principles of person-centred care. Person-centred care improves nursing performance across various clinical settings [28,29,30]. Currently, a paradigm shift is occurring in ICUs—from a survival-focused model to a person-centred approach that emphasises survival, nursing care, and patient respect [24]. Therefore, contemporary research is needed to explore the relationship between person-centred care and delirium nursing performance in TICU nurses.

Thus, this study aims to address the relationships among delirium-related stress, self-efficacy, person-centred care, and delirium nursing performance. Furthermore, we seek to identify the factors influencing delirium nursing performance among TICU nurses (Figure 1).

## 2. Materials and Methods

### 2.1. Study Design

This cross-sectional, descriptive study investigated the effects of delirium-related stress, self-efficacy, and person-centred care on delirium nursing performance in TICUs.

### 2.2. Participants

The study included TICU nurses who had worked for at least 12 months at four general hospitals and four tertiary general hospitals nationwide. The required sample size was calculated using G*Power 3.1.9.7. For multiple regression analysis, a minimum of 166 participants was calculated based on a 0.05 significance level, 0.15 medium effect [31], 90% power, and 14 predictor variables (11 general characteristics and three independent variables). Accounting for a 10% dropout rate, questionnaires were distributed to 185 participants. A total of 180 questionnaires were initially collected. After excluding 10 questionnaires with incomplete responses, 170 were included in the final analysis.

### 2.3. Study Tools

#### 2.3.1. General Characteristics

Participants’ general characteristics were assessed using 11 variables related to general characteristics: sex, age, marital status, educational status, religion, total clinical experience, experience in the TICU, work shift type, nursing guidelines for presence of delirium, use of a delirium assessment tool, and delirium-related education experience.

#### 2.3.2. Delirium-Related Stress

Delirium-related stress was measured with the Delirium Nursing Stress Scale, an instrument developed by Kim [32] and revised by Park [22], used with approval from the author. This 20-item scale is categorised into four subdomains: knowledge of delirium, environment, interpersonal relations, and nursing practice and scope. Scores range from 0 (no stress) to 100 (extreme stress), with higher scores indicating greater levels of stress experienced by nurses caring for patients with delirium. The Cronbach’s α was 0.88 in Park’s original study [22] and 0.88 in the present study.

#### 2.3.3. Self-Efficacy

Self-efficacy was measured using the Self-Efficacy in Clinical Performance instrument, originally developed by Cheraghi et al. [33] for nursing students and translated into Korean by Jung [34], with approval from the author. Each item is rated on a 5-point Likert scale, with higher scores indicating greater self-efficacy. In terms of reliability, the Cronbach’s α was 0.96 in the original study [33] and 0.98 in the study by Jung [34] that included TICU nurses. The Cronbach’s α was 0.96 in this study.

#### 2.3.4. Person-Centred Care

The Person-Centred Nursing Assessment Tool, a validated instrument developed by Lee [35], was used in this study with approval from the author. The tool consists of 25 items divided into five subdomains: cooperative relationship, holism, respect, individualisation, and empowerment. Each item is rated on a five-point Likert scale, with higher scores indicating a higher level of person-centred nursing performance. The reliability of the tool was demonstrated with a Cronbach’s α of 0.94 in the original study [35] and 0.93 in the present study.

#### 2.3.5. Delirium Nursing Performance

The instrument used to measure delirium nursing performance, developed by Yang [12], was used in this study with approval from the author. This tool consists of 25 items, each rated on a five-point Likert scale, where higher scores indicate better performance in delirium nursing care. The Cronbach’s α was 0.84 in the previous study involving TICU nurses [12] and 0.88 in the present study.

### 2.4. Data Collection

After obtaining approval from the Institutional Review Board, data were collected from 22 July to 30 September 2024. The investigator contacted the nursing departments of the eight hospitals operating national TICUs to request their cooperation. For hospitals nearby, the investigator visited the nursing departments in person and explained the purpose and details of the study. For hospitals located farther away, questionnaires were sent by postal mail, and the study’s purpose and procedures were explained over the phone. The study’s purpose and methods were explained to the head nurses, who then informed nurses interested in participating. The head nurses also administered the survey questionnaires. Participants who agreed to take part in the study provided written informed consent and completed the self-administered questionnaires. To protect participant privacy, completed questionnaires were collected in sealed envelopes, and each was assigned a unique serial number instead of a name. As a token of appreciation, participants received a small gift for their participation.

### 2.5. Ethical Considerations

This study was conducted after review and approval by the Institutional Review Board of the authors’ affiliated institution (CBNU-2024-A-0008). Participants received a written information sheet outlining the study’s purpose and procedures and signed a consent form before participation. The sheet explained that participation involved no anticipated risks or direct benefits. It clarified that participants could refuse to complete the questionnaire or withdraw from the study at any point during its completion without incurring any disadvantage or penalty. It also stated that the study results would be used and published solely for research purposes. Collected questionnaires, assigned arbitrary subject identification numbers to ensure anonymity, were stored securely in a locked filing cabinet. The questionnaires were also digitised, stored on a separate mobile device, and protected with a password.

### 2.6. Data Analysis

Data collected in this study were analysed using descriptive statistics, independent *t*-tests, and one-way ANOVA with SPSS Statistics 29.0. The relationship of delirium-related stress, self-efficacy, and person-centred care with delirium nursing performance was assessed using Pearson correlation coefficients. Factors influencing participants’ delirium nursing performance were analysed using the enter method in multiple regression analysis.

## 3. Results

### 3.1. Differences in Delirium Nursing Performance According to Participants’ General Characteristics

The mean age of participants was 29.74 ± 5.81 years. The mean duration of clinical practice was 6.48 ± 5.56 years, and the mean clinical experience in the TICU was 3.93 ± 2.26 years. No significant differences were observed in delirium nursing performance according to general characteristics (Table 1).

### 3.2. Scores of Delirium-Related Stress, Self-Efficacy, Person-Centred Care, and Delirium Nursing Performance

Among TICU nurses, the mean delirium-related stress score was 57.42 ± 13.73 (out of 100 points), the mean self-efficacy score was 3.82 ± 0.40 (out of five points), and the mean person-centred care score was 3.85 ± 0.39 (out of five points). Among the subdomains of person-centred care, the highest-scoring item was respect (4.03 ± 0.49), followed by individualisation (4.01 ± 0.43), cooperative relationship (3.82 ± 0.43), empowerment (3.68 ± 0.58), and holism (3.67 ± 0.53). The mean delirium nursing performance score was 3.80 ± 0.44 out of five points (Table 2).

### 3.3. Correlation of Delirium-Related Stress, Self-Efficacy, and Person-Centred Care with Delirium Nursing Performance in Participants

Delirium nursing performance showed significant positive correlations with self-efficacy (*r* = 0.41, *p* < 0.001) and person-centred care (*r* = 0.51, *p* < 0.001). Additionally, positive correlations were found between delirium nursing performance and the subdomains of person-centred care, including cooperative relationship (*r* = 0.32, *p* < 0.001), holism (*r* = 0.37, *p* < 0.001), respect (*r* = 0.44, *p* < 0.001), individualisation (*r* = 0.46, *p* < 0.001), and empowerment (*r* = 0.49, *p* < 0.001) (Table 3).

### 3.4. Factors Influencing Delirium Nursing Performance

Person-centred care and self-efficacy, significantly correlated with delirium nursing performance, were included in the regression analysis as continuous variables. These variables were entered into a multiple regression model. The Durbin–Watson statistic was 1.896, which is close to 2, indicating no autocorrelation among the independent variables. Tolerance level ranged from 0.41 to 0.51, all above the threshold of 0.1, and the variance inflation factor was 1.28, below the threshold of 10, indicating that multicollinearity was not an issue. The regression analysis revealed that person-centred care (β = 0.41, *p* < 0.001) and self-efficacy (β = 0.22, *p* = 0.004) were significant predictors of delirium nursing performance. The regression model for delirium nursing performance was statistically significant (F = 34.82, *p* < 0.001), with an explanatory power of 28.6% (Table 4).

## 4. Discussion

This study examined the relationship of delirium-related stress, self-efficacy, and person-centred care with delirium nursing performance and identified factors influencing delirium nursing performance among TICU nurses. Person-centred care and self-efficacy significantly influenced delirium nursing performance among TICU nurses, explaining 28.6% of the variance.

Person-centred care was the most critical factor influencing delirium nursing performance among TICU nurses. The findings of this study align with the results of previous research demonstrating the positive impact of person-centred care on nursing practice [28,29]. Moreover, clinical nurses involved in patient safety management identified person-centred care as the only influential factor, rather than clinical competency [28]. Another study also reported that person-centred care significantly affected patient safety practices among operating room nurses [29]. These findings highlight the importance of holistic nursing care that integrates emotional and religious aspects with physical healthcare, particularly for patients in the acute phase of illness. Especially, communication between nurses and patients in the TICU is often limited due to brain injuries or multiple traumas. In most cases, patients and their families are excluded from the medical decision-making processes [21]. However, ICU nurses providing effective, communication-based family support during the family visiting hours improved the quality of patient care [36]. Active nursing care through effective communication, beginning with recognising values of person-centred care, would be a helpful strategy to effectively perform nursing care for patients with delirium in TICUs, along with other strategies that improve person-centred care.

Person-centred care scores among TICU nurses in this study were lower than those previously reported among clinical nurses [28,36]. However, the scores were comparable to those observed among ICU nurses [21,24,29] and nurses caring for post-operative patients [29]. Notably, the holism subscale received the lowest score, consistent with the findings from studies using the same instrument [28,37]. This suggests that because TICU nurses prioritise addressing the patients’ critical physical conditions and pathophysiology, they may find it challenging to provide person-centred care that includes holistic needs. Nevertheless, given that this study identified person-centred care as the most significant factor in improving the quality of delirium nursing care, interventions aimed at enhancing nurses’ awareness of holism and empowerment—the two lowest-scoring domains—should be implemented to strengthen their capacity to deliver comprehensive nursing care.

Self-efficacy was the second most influential factor affecting nursing performance in TICUs. This finding aligns with the results of previous studies that investigated the impact of self-efficacy on delirium nursing performance in various nursing care settings. In a previous study on hospice nurses, those with high self-efficacy in nursing practice demonstrated better performance in caring for patients with delirium [38]. Similarly, self-efficacy was associated with delirium nursing performance among nurses providing care in recovery rooms [18] and general wards [39]. ICU nurses also reported that self-efficacy has a significant impact on job performance [40]. Nurses with high self-efficacy are more likely to rapidly adapt to changes in patients’ conditions and apply appropriate nursing interventions when delirium symptoms arise [19]. Self-efficacy among ICU nurses can be improved through delirium-related education [17], as it is influenced by experience and verbal persuasion, both of which can be addressed through educational programs [11]. One study investigating the effects of a delirium-focused educational program on ICU nurses’ self-efficacy showed improvements in delirium-related nursing practices [11]. Therefore, delirium-focused educational programs that include content aimed at improving self-efficacy should be provided to nurses caring for patients with delirium, particularly those with multiple injuries and severe brain injuries where delirium frequently occurs.

The level of delirium-related stress in this study was similar to that reported among recovery room nurses; however, the maximum score recorded was higher (93.0 vs. 87.5) [18]. ICU nurses who frequently provide care for patients with delirium often experience serious safety risks and emotional distress [14,16]. Moreover, delirium-related stress showed a significant negative correlation with delirium nursing performance in this study, consistent with findings from a previous study [15]. Managing delirium could lead to psychological burden, manifested as delirium-related stress [15]. However, no effect of delirium-related stress on delirium nursing performance was observed in this study. Similar findings were reported in studies involving recovery room nurses [15,18]. Therefore, further research is warranted to explore the effect of delirium-related stress on delirium nursing performance in ICU nurses. Although the effect of delirium-related stress on delirium nursing performance, based on the correlation between these variables, the study suggested that nursing strategies aimed at improving coping skills and psychological well-being are necessary, as these experiences can contribute to emotional stress.

This study has some limitations. The findings cannot be generalised to the broader population, as the study included only TICU nurses in Korea. Additionally, this cross-sectional study did not assess long-term changes; thus, we could not determine causality. Moreover, the nurses who were not present during the survey did not participate. TICUs in Korea are operated at a national level, potentially introducing cultural limitations. Furthermore, the use of self-reported questionnaires instead of objective measures may have introduced subjective bias, including social desirability or inaccurate self-assessment. These issues may have affected the accuracy of the data, including the results on delirium-related stress, self-efficacy, person-centred care, and delirium nursing performance. Therefore, further studies should employ a broad-scale, randomised sampling to obtain more representative data.

Nevertheless, this study highlights the importance of managing delirium in TICUs, where its incidence is high, and underscores the significant impact of person-centred care and self-efficacy in performing nursing interventions on delirium nursing performance. These findings provide valuable insights for improving delirium nursing practices and contribute essential data to the field of TICU nursing care.

## 5. Conclusions

In this study, person-centred care was the most critical factor influencing delirium nursing performance among TICU nurses. Moreover, self-efficacy significantly contributed to delirium nursing performance. According to the study findings, the development of educational programs focused on person-centred care may improve delirium nursing performance and support holistic care practices. Systemic changes within healthcare organisations should also be implemented to create a work environment where nurses have the time and opportunity to listen closely to patients. Furthermore, healthcare institutions and nurse managers should recognise and address psychological problems, such as delirium-related stress, by providing appropriate support for TICU nurses. Future research should focus on developing various simulation-based educational programs related to delirium nursing care to enhance person-centred care and improve self-efficacy. Additionally, studies are needed to evaluate the impact of delirium-related stress on delirium nursing performance in TICU to improve the quality of TICU nurses’ psychological well-being. This study serves as a foundational resource for the development of educational and interventional programs that improve person-centred care and enhance self-efficacy in delirium nursing performance.

## Figures and Tables

**Figure 1 healthcare-13-01243-f001:**
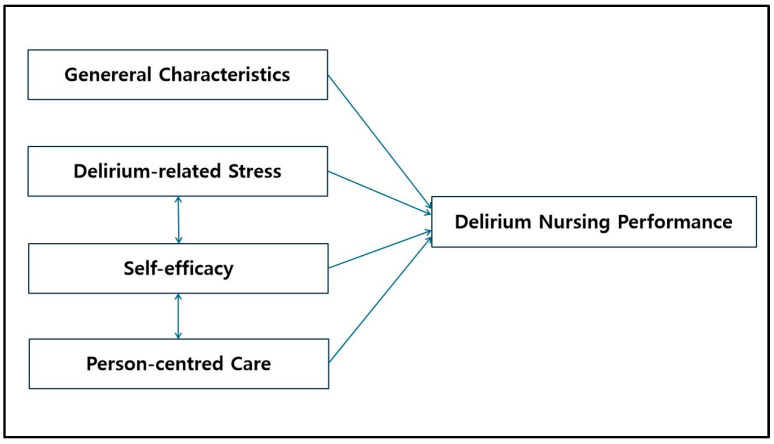
Conceptual framework of this study.

**Table 1 healthcare-13-01243-t001:** Differences in delirium nursing performance according to general characteristics (*N* = 170).

Variables	Categories	*n*	%	Delirium Nursing Performance	
				M ± SD	*t*/F (*p*)
Sex	Female	134	78.8	3.80 ± 0.42	0.05 (0.957)
	Male	36	21.2	3.80 ± 0.50	
Age (years)	<30	107	62.9	3.80 ± 0.44	0.01 (0.994)
	30∼40	51	30.0	3.80 ± 0.46	
	≥40	12	7.1	3.79 ± 0.42	
Marital status	Unmarried	134	76.1	3.79 ± 0.44	−0.72 (0.473)
	Married	42	23.9	3.86 ± 0.44	
Educational status	≤Bachelors	153	90.0	3.79 ± 0.43	−0.78 (0.439)
	≥Masters	17	10.0	3.87 ± 0.55	
Religion	No	131	77.1	3.78 ± 0.42	−0.84 (0.400)
	Yes	39	22.9	3.85 ± 0.50	
Total clinical experience (years)	<3	46	27.1	3.78 ± 0.48	0.59 (0.625)
	3∼5	37	21.8	3.85 ± 0.43	
	5∼10	57	33.5	3.82 ± 0.45	
	≥10	30	17.6	3.72 ± 0.39	
Experience in the trauma intensive care unit (years)	<3	68	40.0	3.76 ± 0.48	0.72 (0.488)
	3∼5	53	31.2	3.86 ± 0.40	
	≥5	49	28.8	3.79 ± 0.43	
Work shift type	3 shifts	162	95.3	3.81 ± 0.44	1.08 (0.281)
	2 shifts, full-time	8	4.7	3.64 ± 0.37	
Presence of nursing guidelines for delirium care	Yes	65	38.2	3.87 ± 0.42	1.59 (0.113)
	None	105	61.8	3.75 ± 0.45	
Use of a delirium assessment tool	Used	94	55.3	3.84 ± 0.46	1.51 (0.133)
	Not used	76	44.7	3.74 ± 0.41	
Delirium-related education experience	Yes	103	60.6	3.83 ± 0.43	1.29 (0.198)
	No	67	39.4	3.75 ± 0.45	

Notes. *n*: frequency; M: mean; SD: standard deviation.

**Table 2 healthcare-13-01243-t002:** Level of delirium-related stress, self-efficacy, person-centred care, and delirium nursing performance.

Variables	Number of Items	Mean			Scale Range
		M ± SD	Min	Max	
Delirium-related stress	20	57.42 ± 13.73	12.55	93.00	0–100
Self-efficacy	37	3.82 ± 0.40	2.84	5.00	1–5
Person-centred care	25	3.85 ± 0.39	2.84	5.00	1–5
Cooperative relationship	7	3.82 ± 0.43	2.86	5.00	1–5
Holism	4	3.67 ± 0.53	2.25	5.00	1–5
Respect	5	4.03 ± 0.49	2.80	5.00	1–5
Individualisation	5	4.01 ± 0.43	2.80	5.00	1–5
Empowerment	4	3.68 ± 0.58	2.25	5.00	1–5
Delirium nursing performance	25	3.80 ± 0.44	2.80	4.88	1–5

Notes. M: mean; SD: standard deviation; Min: minimum; Max: maximum.

**Table 3 healthcare-13-01243-t003:** Correlation of delirium-related stress, person-centred care, and self-efficacy with delirium nursing performance (*N* = 170).

Variables	1. DS	2. SE	3. PCC						4. DNP
			Total	3.1	3.2	3.3	3.4	3.5	
	*r* (*p*)								
1. DS	1								
2. SE	0.12 (0.112)	1							
3. PCC	−0.05 (0.535)	0.47 (<0.001)	1						
3.1. Cooperative relationship	−0.12 (0.108)	0.37 (<0.001)	0.84 (<0.001)	1					
3.2. Holism	0.03 (0.731)	0.32 (<0.001)	0.71 (<0.001)	0.51 (<0.001)	1				
3.3. Respect	−0.08 (0.318)	0.37 (<0.001)	0.85 (<0.001)	0.65 (<0.001)	0.49 (<0.001)	1			
3.4. Individualisation	0.08 (0.302)	0.48 (<0.001)	0.84 (<0.001)	0.63 (<0.001)	0.47 (<0.001)	0.71 (<0.001)	1		
3.5. Empowerment	0.06 (0.461)	0.33 (<0.001)	0.77 (<0.001)	0.50 (<0.001)	0.44 (<0.001)	0.57 (<0.001)	0.60 (<0.001)	1	
4. DNP	−0.01 (0.905)	0.41 (<0.001)	0.51 (<0.001)	0.32 (<0.001)	0.37 (<0.001)	0.44 (<0.001)	0.46 (<0.001)	0.49 (<0.001)	1

Notes. DS: delirium-related stress; SE: self-efficacy; PCC: person-centred care; DNP: delirium nursing performance.

**Table 4 healthcare-13-01243-t004:** Factors influencing delirium nursing performance (*N* = 170).

	B	SE	β	*t* (*p*)
Constant	1.11	0.33		3.41 (<0.001)
Person-centred care	0.46	0.08	0.41	5.53 (<0.001)
Self-Efficacy	0.24	0.08	0.22	2.96 (0.004)
F (*p*)	34.82 (<0.001)			
R^2^ (Adjusted R^2^)	0.294 (0.286)			
Tolerance	0.41–0.51			
Variance inflation factor	1.28			

Notes. B: unstandardised regression coefficient; SE: standard error; β: standardised regression coefficient.

## Data Availability

Data are included within the article.

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
