# Peer review of "Impact of Delirium-Related Stress, Self-Efficacy, Person-Centred Care on Delirium Nursing Performance Among Nurses in Trauma Intensive Care Units: A Cross-Sectional Descriptive Survey Study"

_healthcare, 2025, doi:10.3390/healthcare13111243_

Round 1

Reviewer 1 Report

Comments and Suggestions for Authors

In my opinion, the objective of the study is clearly defined, and the methodology is appropriate and well-executed. The discussion is coherent, well-structured, and appropriately supported by the results and existing literature.

Overall, I find your work to be relevant and of high quality, and I believe it makes a meaningful contribution to the field.

Updated:

The introduction successfully contextualizes the research problem, justifies its relevance, and articulates the study’s objectives with precision. It is further supported by a current and pertinent review of the literature, which demonstrates a solid understanding of the state of the art in the field.

The methodology section is described with clarity and rigor. It specifies the study design, inclusion and exclusion criteria, sample characteristics, variables under consideration, and the procedures for data collection and analysis.

The selected sample is appropriate and well-defined. The sampling strategies and the rationale for their selection are clearly articulated, which reinforces the study’s internal validity.

The results are presented in a structured, clear, and coherent manner, both within the text and in the accompanying tables and figures. The tables are well-designed, enhance the clarity of the findings, and effectively support the narrative presentation of the results. The statistical analyses are appropriate, and the data are interpreted with due caution and precision, avoiding any overinterpretation.

The discussion effectively relates the findings to the existing body of scientific literature. The study’s key contributions are clearly emphasized, and its limitations are acknowledged in a transparent manner, thereby strengthening the critical appraisal.

The conclusions are clearly stated and aligned with the study’s findings. Their relevance to clinical practice is duly noted, highlighting the practical implications of the research.

The manuscript also addresses the necessary ethical considerations, ensuring adherence to the principles of scientific integrity. With regard to the references, the sources cited are current, relevant, and properly formatted in accordance with the journal’s required citation style. In light of the above, I believe the manuscript offers an original and meaningful contribution to its field, meets the criteria of scientific rigor, and merits consideration for publication.

Author Response

Response to Reviewer 1 Comments

My co-author and I wish to resubmit our revised manuscript entitled “Impact of Delirium-related Stress, Self-Efficacy, Person-Centred Care on Delirium Nursing Performance Among Nurses in Trauma Intensive Care Units: A Cross-sectional Descriptive Survey Study” with changes that thoroughly address your comments.

We thank you for your thoughtful suggestions and insights. The manuscript has benefited from your thorough feedback. We look forward to working with you to move this manuscript closer to publication in Healthcare.

The manuscript has been rechecked, and the necessary changes have been made in accordance with your suggestions. The responses to all comments have been prepared and attached herewith/given below.

We appreciate your consideration.

Comment 1: The introduction successfully contextualizes the research problem, justifies its relevance, and articulates the study’s objectives with precision. It is further supported by a current and pertinent review of the literature, which demonstrates a solid understanding of the state of the art in the field.

Response 1: Thank you for valuable feedback. In response, we revised the final section of the Introduction to present the research question. Additionally, following Reviewer 3’s suggestion, we incorporated a conceptual framework (Figure 1) to illustrate the hypothesised relationships among variables. Moreover, 70% of the references were cited from papers published within the past 5 years. Older sources were used only for the definition of key concepts and descriptions of the measurement tools (Page 3, Paragraph 2, Lines 102–108).

“Thus, this study aims to address the relationships among delirium-related stress, self-efficacy, person-centred care, and delirium nursing performance. Furthermore, we seek to identify the factors influencing delirium nursing performance among TICU nurses (Figure 1).

Figure 1. Conceptual framework of this study”

Comment 2: The methodology section is described with clarity and rigor. It specifies the study design, inclusion and exclusion criteria, sample characteristics, variables under consideration, and the procedures for data collection and analysis.

Response 2: Thank you for your constructive feedback. Our manuscript included a description of the study design, participant inclusion criteria and general characteristics, measurement instruments, data collection and analysis, and ethical considerations.

Comment 3: The selected sample is appropriate and well-defined. The sampling strategies and the rationale for their selection are clearly articulated, which reinforces the study’s internal validity.

Response 3: We calculated the sample size using G*power for multiple regression (Page 3, Paragraph 4, Lines 115–123)

“2.2. Participants

The study included TICU nurses who had worked for at least 12 months at four general hospitals and four tertiary general hospitals nationwide. The required sample size was calculated using G*Power 3.1.9.7. For multiple regression analysis, a minimum of 166 participants was calculated based on a 0.05 significance level, 0.15 medium effect [31], 90% power, and 14 predictor variables (11 general characteristics and 3 independent variables). Accounting for a 10% dropout rate, questionnaires were distributed to 185 participants. A total of 180 questionnaires were initially collected. After excluding 10 questionnaires with incomplete responses, 170 were included in the final analysis.”

Comment 4: The results are presented in a structured, clear, and coherent manner, both within the text and in the accompanying tables and figures. The tables are well-designed, enhance the clarity of the findings, and effectively support the narrative presentation of the results. The statistical analyses are appropriate, and the data are interpreted with due caution and precision, avoiding any overinterpretation.

Response 4: Thank you for your positive feedback. The results are summarised in tables according to their description in the results section. The results in the tables are briefly referred to in the text.

Comment 5: The discussion effectively relates the findings to the existing body of scientific literature. The study’s key contributions are clearly emphasized, and its limitations are acknowledged in a transparent manner, thereby strengthening the critical appraisal.

Response 5: Thank you for your thoughtful feedback. We discussed the main findings of the study based on the cited literature. We particularly addressed the discrepancy between the significant correlation and the non-significant regression result for delirium-related stress (Page 9, Paragraph 2, Lines 304–317). Moreover, we discussed the study’s limitations, including the cross-sectional study design and self-report data collection method (Page 9, Paragraph 3, Lines 319–328).

“The level of delirium-related stress in this study was similar to that reported among recovery room nurses; however, the maximum score recorded was higher (93.0 vs. 87.5) [18]. ICU nurses who frequently provide care for patients with delirium often experience serious safety risks and emotional distress [14,16]. Moreover, delirium-related stress showed a significant negative correlation with delirium nursing performance in this study, consistent with findings from a previous study [15]. Managing delirium could lead to psychological burden, manifested as delirium-related stress [15]. However, no effect of delirium-related stress on delirium nursing performance was observed in this study. Similar findings were reported in studies involving recovery room nurses [15,18]. Therefore, further research is warranted to explore the effect of delirium-related stress on delirium nursing performance in ICU nurses. Although the effect of delirium-related stress on delirium nursing performance, based on the correlation between these variables, the study suggested that nursing strategies aimed at improving coping skills and psychological well-being are necessary, as these experiences can contribute to emotional stress.”

“Additionally, this cross-sectional study did not assess long-term changes; thus, we could not determine causality. Moreover, the nurses who were not present during the survey did not participate. TICUs in Korea are operated at a national level, potentially introducing cultural limitations. Furthermore, the use of self-reported questionnaires instead of objective measures may have introduced subjective bias, including social desirability or inaccurate self-assessment. These issues may have affected the accuracy of the data, including the results on delirium-related stress, self-efficacy, person-centred care, and delirium nursing performance. Therefore, further studies should employ a broad-scale, randomised sampling to obtain more representative data.”

Comment 6: The conclusions are clearly stated and aligned with the study’s findings. Their relevance to clinical practice is duly noted, highlighting the practical implications of the research.

Response 6: Thank you for your valuable feedback. We summarized the results of the study in the conclusion and emphasised its practical implications. We also added suggestions at the institutional level and provided recommendations for future research (Pages 9-10, Paragraph 1, Lines 340–349).

“Systemic changes within healthcare organisations should also be implemented to create a work environment where nurses have the time and opportunity to listen closely to patients. Furthermore, healthcare institutions and nurse managers should recognise and address psychological problems, such as delirium-related stress, by providing appropriate support for TICU nurses. Future research should focus on developing various simulation-based educational programs related to delirium nursing care to enhance person-centred care and improve self-efficacy. Additionally, studies are needed to evaluate the impact of delirium-related stress on delirium nursing performance in TICU to improve the quality of TICU nurses’ psychological well-being.”

Comment 7: The manuscript also addresses the necessary ethical considerations, ensuring adherence to the principles of scientific integrity. With regard to the references, the sources cited are current, relevant, and properly formatted in accordance with the journal’s required citation style.

Response 7: Thank you for your thoughtful comment. We have described the ethical considerations related to our study in the Methods section (Page 5, Paragraph 1, Lines 178–189). Regarding references, 70% of cited sources were published within the last 5 years. All references have been formatted according to the journal’s required citation style.

“2.5. Ethical considerations

This study was conducted after review and approval by the Institutional Review Board of the authors’ affiliated institution (CBNU-2024-A-0008). Participants received a written information sheet outlining the study’s purpose and procedures and signed a consent form before participation. The sheet explained that participation involved no anticipated risks or direct benefits. It clarified that participants could refuse to complete the questionnaire or withdraw from the study at any point during its completion without incurring any disadvantage or penalty. It also stated that the study results would be used and published solely for research purposes. Collected questionnaires, assigned arbitrary subject identification numbers to ensure anonymity, were stored securely in a locked filing cabinet. The questionnaires were also digitised, stored on a separate mobile device, and protected with a password.”

Comment 8: In light of the above, I believe the manuscript offers an original and meaningful contribution to its field, meets the criteria of scientific rigor, and merits consideration for publication.

Response 8: We deeply appreciate your thoughtful review.

END OF RESPONSES TO COMMENTS

Please let me know if you have any other concerns or questions about it. Thank you.

Reviewer 2 Report

Comments and Suggestions for Authors

This is a well-structured article, with a solid theoretical foundation and rigorous methodology. The approach is relevant, and the focus on the role of self-efficacy and person-centred care contributes significantly to improving care in trauma intensive care units.

Abstract
It should include a conceptual definition of the central themes of the article: “Delirium-related Stress,” “Self-Efficacy,” “Person-Centred Care,” “Delirium Nursing Performance Among Nurses,” and “Trauma Intensive Care Unit.”
The methods section should specify how data collection was carried out.

Introduction
It provides a comprehensive overview of the problem of delirium in TICUs, supported by relevant literature. The choice of the three constructs—stress, self-efficacy, and person-centred care—is well justified.
As a suggestion for improvement, it could end with a clear research statement such as: “Thus, this study aims to address the following question...”.

Discussion
The critique regarding the limitations of the instruments (subjective self-assessment) and the cross-sectional design could be further developed.
Delirium-related stress showed no correlation with performance, yet this result is only superficially discussed.

Conclusion
The article lacks concrete suggestions regarding the practical implications of the study at the level of institutional strategies.
It should also include recommendations for future research or studies.

Author Response

Response to Reviewer 2 Comments

My co-author and I wish to resubmit our revised manuscript entitled “Impact of Delirium-related Stress, Self-Efficacy, Person-Centred Care on Delirium Nursing Performance Among Nurses in Trauma Intensive Care Units: A Cross-sectional Descriptive Survey Study” with changes that thoroughly address your comments.

We thank you for your thoughtful suggestions and insights. The manuscript has benefited from your thorough feedback. We look forward to working with you to move this manuscript closer to publication in Healthcare.

The manuscript has been rechecked, and the necessary changes have been made in accordance with your suggestions. The responses to all comments have been prepared and attached herewith/given below.

We appreciate your consideration.

Comment 1: Abstract_It should include a conceptual definition of the central themes of the article: “Delirium-related Stress,” “Self-Efficacy,” “Person-Centred Care,” “Delirium Nursing Performance Among Nurses,” and “Trauma Intensive Care Unit.”. The methods section should specify how data collection was carried out.

Response 1: Thank you for your valuable recommendations. We added the description about the conceptual definition of the central themes and data collection method (Page 1, Abstract, Lines 12-18, 22–23).

“Enhancing delirium nursing performance in trauma intensive care units (TICUs), where the prevalence of delirium is high, causes early detection of delirium and improves the quality of nursing care. TICU nurses experience various stress levels while caring for patients with delirium, which negatively affects their performance. Self-efficacy improves delirium nursing performance based on their capacity. Person-centred care identifies the holistic needs of patients in TICUs, which stimulates their recovery. This study aimed to examine the relationship of delirium-related stress, self-“

“Data was collected using self-reported questionnaires after informed consent was provided. Data were analysed using multiple”

Comment 2: Introduction_It provides a comprehensive overview of the problem of delirium in TICUs, supported by relevant literature. The choice of the three constructs—stress, self-efficacy, and person-centred care—is well justified. As a suggestion for improvement, it could end with a clear research statement such as: “Thus, this study aims to address the following question...”.

Response 2: Thank you for your kind advice. Following your suggestion, we revised the research statement as follows (Page 3, Paragraph 2, Lines 102–105).

“Thus, this study aims to address the relationships among delirium-related stress, self-efficacy, person-centred care, and delirium nursing performance. Furthermore, we seek to identify the factors influencing delirium nursing performance among TICU nurses (Figure 1).”

Comment 3: Discussion_The critique regarding the limitations of the instruments (subjective self-assessment) and the cross-sectional design could be further developed.

Response 3: Following your detailed suggestions, we further expanded that section to include the limitations introduced by the self-report assessment method and cross-sectional study design (Page 9, Paragraph 3, Lines 318–328).

“This study has some limitations. The findings cannot be generalised to the broader population, as the study included only TICU nurses in Korea. Additionally, this cross-sectional study did not assess long-term changes; thus, we could not determine causality. Moreover, the nurses who were not present during the survey did not participate. TICUs in Korea are operated at a national level, potentially introducing cultural limitations. Furthermore, the use of self-reported questionnaires instead of objective measures may have introduced subjective bias, including social desirability or inaccurate self-assessment. These issues may have affected the accuracy of the data, including the results on delirium-related stress, self-efficacy, person-centred care, and delirium nursing performance. Therefore, further studies should employ a broad-scale, randomised sampling to obtain more representative data.”

Comment 4: Discussion_Delirium-related stress showed no correlation with performance, yet this result is only superficially discussed.

Response 4: Thank you for your insightful feedback. We revised the discussion section and clarified the level of delirium-related stress and its negative correlation with nursing performance, along with regression analysis results (Pages 9, Paragraph 2, lines 304–317).

“The level of delirium-related stress in this study was similar to that reported among recovery room nurses; however, the maximum score recorded was higher (93.0 vs. 87.5) [18]. ICU nurses who frequently provide care for patients with delirium often experience serious safety risks and emotional distress [14,16]. Moreover, delirium-related stress showed a significant negative correlation with delirium nursing performance in this study, consistent with findings from a previous study [15]. Managing delirium could lead to psychological burden, manifested as delirium-related stress [15]. However, no effect of delirium-related stress on delirium nursing performance was observed in this study. Similar findings were reported in studies involving recovery room nurses [15,18]. Therefore, further research is warranted to explore the effect of delirium-related stress on delirium nursing performance in ICU nurses. Although the effect of delirium-related stress on delirium nursing performance, based on the correlation between these variables, the study suggested that nursing strategies aimed at improving coping skills and psychological well-being are necessary, as these experiences can contribute to emotional stress.”

Comment 5: Conclusion_The article lacks concrete suggestions regarding the practical implications of the study at the level of institutional strategies. It should also include recommendations for future research or studies.

Response 5: Thank you for your valuable recommendation. In response, we have revised the conclusion to include practical implications at the institutional level, as well as recommendations for future research (Pages 9-10, Paragraph 5-1, Lines 340–349).

“Systemic changes within healthcare organisations should also be implemented to create a work environment where nurses have the time and opportunity to listen closely to patients. Furthermore, healthcare institutions and nurse managers should recognise and address psychological problems, such as delirium-related stress, by providing appropriate support for TICU nurses. Future research should focus on developing various simulation-based educational programs related to delirium nursing care to enhance person-centred care and improve self-efficacy. Additionally, studies are needed to evaluate the impact of delirium-related stress on delirium nursing performance in TICU to improve the quality of TICU nurses’ psychological well-being.”

END OF RESPONSES TO COMMENTS

Please let me know if you have any other concerns or questions about it. Thank you.

Reviewer 3 Report

Comments and Suggestions for Authors

This manuscript addresses an important gap in trauma intensive care unit (TICU) nursing by examining how self-efficacy and person-centred care relate to delirium nursing performance (which is an ambiguous term!).

Page 1, Line 12: Consider revising “This aimed to examine...” to “This study aimed to examine...”.
Page 1, Line 14: “Delirium nursing performance” is repeatedly used but not clearly defined. A brief operational definition early in the text would improve clarity. This is also applicable for "Delirium-related Stress". Please define all variables in the introduction. 

Page 1, Line 12: Consider revising “This aimed to examine...” to “This study aimed to examine...”.

Page 1, Line 14: The term “delirium nursing performance” is frequently used but not clearly defined. Similarly, “delirium-related stress” is introduced without an operational definition. For clarity and coherence, we strongly encourage defining all key variables explicitly in the Introduction section, ideally supported by citations where available.

To further enhance reader understanding, especially for interdisciplinary or international audiences, consider adding a conceptual figure early in the manuscript (e.g., at the end of the Introduction). This figure could map out the hypothesized relationships among variables (e.g., self-efficacy, person-centred care, delirium-related stress, and nursing performance). A visual model will help anchor the study's conceptual foundation and strengthen the clarity of the research framework.

Author Response

Response to Reviewer 3 Comments

My co-author and I wish to resubmit our revised manuscript entitled “Impact of Delirium-related Stress, Self-Efficacy, Person-Centred Care on Delirium Nursing Performance Among Nurses in Trauma Intensive Care Units: A Cross-sectional Descriptive Survey Study” with changes that thoroughly address your comments.

We thank you for your thoughtful suggestions and insights. The manuscript has benefited from your thorough feedback. We look forward to working with you to move this manuscript closer to publication in Healthcare.

The manuscript has been rechecked, and the necessary changes have been made in accordance with your suggestions. The responses to all comments have been prepared and attached herewith/given below.

We appreciate your consideration.

Comment 1: Page 1, Line 12: Consider revising “This aimed to examine...” to “This study aimed to examine...”.

Response 1: Thank you for your kind recommendation. We revised this sentence (Page 1, Abstract, Line 18).

“This study aimed to examine the relationship of delirium-related stress, self-efficacy, and person-centred care with delirium nursing performance and identify factors”

Comment 2: Page 1, Line 14: “Delirium nursing performance” is repeatedly used but not clearly defined. A brief operational definition early in the text would improve clarity. This is also applicable for "Delirium-related Stress". Please define all variables in the introduction. 

Page 1, Line 14: The term “delirium nursing performance” is frequently used but not clearly defined. Similarly, “delirium-related stress” is introduced without an operational definition. For clarity and coherence, we strongly encourage defining all key variables explicitly in the Introduction section, ideally supported by citations where available.

Response 2: Thank you for your meaningful advice. Following your suggestion, we revised the introduction and included definitions of all key variables (Pages 2-3, Lines 54, 63-64, 75–76, 91–95).

“Prompt identification of delirium-related risks through vigilant care, defined as delirium nursing performance [12], is crucial for effective intervention.”

“Delirium-related stress is experienced by nurses during their care of patients with delirium [22].”

“Self-efficacy refers to nurses’ belief in their ability to perform care independently [11,17].”

“Person-centred care is a holistic nursing approach focusing on the individual needs of patients, respecting their personal choices, and protecting their autonomy and dignity [24,25]. According to the conceptual framework of person-centred care, effective care delivery requires that the healthcare system must understand and meet patients' needs [27].”

Comment 3: To further enhance reader understanding, especially for interdisciplinary or international audiences, consider adding a conceptual figure early in the manuscript (e.g., at the end of the Introduction). This figure could map out the hypothesized relationships among variables (e.g., self-efficacy, person-centred care, delirium-related stress, and nursing performance). A visual model will help anchor the study's conceptual foundation and strengthen the clarity of the research framework.

Response 3: Thank you for your valuable recommendation. We added the conceptual framework for this study at the end of the Introduction (Page 3, Figure 1, Lines 102–108).

“Thus, this study aims to address the relationships among delirium-related stress, self-efficacy, person-centred care, and delirium nursing performance. Furthermore, we seek to identify the factors influencing delirium nursing performance among TICU nurses (Figure 1).

Figure 1. Conceptual framework of this study”

END OF RESPONSES TO COMMENTS

Please let me know if you have any other concerns or questions about it. Thank you.

Round 2

Reviewer 3 Report

Comments and Suggestions for Authors

The authors have satisfactorily responded to my comments